# Putative Role of Protein Palmitoylation in Cardiac Lipid-Induced Insulin Resistance

**DOI:** 10.3390/ijms21249438

**Published:** 2020-12-11

**Authors:** Francesco Schianchi, Jan F. C. Glatz, Artur Navarro Gascon, Miranda Nabben, Dietbert Neumann, Joost J. F. P. Luiken

**Affiliations:** 1Department of Genetics & Cell Biology, Faculty of Health, Medicine and Life Sciences, Maastricht University, 6200 MD Maastricht, The Netherlands; f.schianchi@maastrichtuniversity.nl (F.S.); glatz@maastrichtuniversity.nl (J.F.C.G.); a.navarrogascon@student.maastrichtuniversity.nl (A.N.G.); m.nabben@maastrichtuniversity.nl (M.N.); 2Department of Clinical Genetics, Maastricht University Medical Center+, 6202 AZ Maastricht, The Netherlands; 3Department of Pathology, Maastricht University Medical Center+, 6202 AZ Maastricht, The Netherlands; d.neumann@maastrichtuniversity.nl

**Keywords:** palmitoylation, insulin signaling, insulin resistance, cardiac muscle

## Abstract

In the heart, inhibition of the insulin cascade following lipid overload is strongly associated with contractile dysfunction. The translocation of fatty acid transporter CD36 (SR-B2) from intracellular stores to the cell surface is a hallmark event in the lipid-overloaded heart, feeding forward to intracellular lipid accumulation. Yet, the molecular mechanisms by which intracellularly arrived lipids induce insulin resistance is ill-understood. Bioactive lipid metabolites (diacyl-glycerols, ceramides) are contributing factors but fail to correlate with the degree of cardiac insulin resistance in diabetic humans. This leaves room for other lipid-induced mechanisms involved in lipid-induced insulin resistance, including protein palmitoylation. Protein palmitoylation encompasses the reversible covalent attachment of palmitate moieties to cysteine residues and is governed by protein acyl-transferases and thioesterases. The function of palmitoylation is to provide proteins with proper spatiotemporal localization, thereby securing the correct unwinding of signaling pathways. In this review, we provide examples of palmitoylations of individual signaling proteins to discuss the emerging role of protein palmitoylation as a modulator of the insulin signaling cascade. Second, we speculate how protein hyper-palmitoylations (including that of CD36), as they occur during lipid oversupply, may lead to insulin resistance. Finally, we conclude that the protein palmitoylation machinery may offer novel targets to fight lipid-induced cardiomyopathy.

## 1. Introduction

Chronic lipid overload of the heart, as occurs, for instance, during obesity, is known to have major consequences for cardiac functioning and ultimately may elicit cardiac disease. Specifically, excess lipids cause insulin resistance, often leading to cardiac contractile dysfunction. Given the increase in the prevalence of obesity during the last decades, cardiac lipid-induced insulin resistance has become a key contributor to the development of cardiovascular diseases [1,2,3]. Insulin signaling in the heart is important as it represents a major trigger for stimulation of the uptake of long-chain fatty acids (LCFAs) and of glucose, the predominant energy substrates for cardiac muscle [4]. The molecular mechanism underlying lipid-induced insulin resistance is rather complex yet has been elucidated in much detail in the past few years. Still, a number of regulatory events involved in insulin signaling have remained elusive. One of these is the putative role of post-translational modifications (PTMs) of proteins and enzymes in this signaling cascade, such as protein palmitoylation.

Palmitoylation (or S-acylation) refers to the reversible chemical linkage of a palmitate molecule to cysteine residues [5]. S-acylation can influence protein localization, which, in turn, affects its functioning and protein–protein interaction. Unlike other kinds of protein lipidation, such as myristoylation and prenylation that are irreversible modifications [6,7,8], palmitoylation occurs in a dynamic and reversible fashion. Such features of palmitoylation provide proteins with proper synchronicity and spatiotemporal localization [9]. Moreover, because S-acylation is reversibly regulated by enzymes, any malfunctioning of those enzymes can alter palmitoylation patterns, finally compromising protein functioning and signal transduction pathways [10,11]. As a result, palmitoylation, rather than any other kind of fatty acid modification, is considered the major protein lipidation involved in the regulation of signaling cascades. General aspects of the function and regulation of protein palmitoylation have been reviewed previously [12,13,14,15].

In this review, we first describe the insulin signaling pathway in the heart (Section 2.1) and evaluate both current knowledge and discrepancies concerning the molecular mechanisms leading to lipid-induced insulin resistance (Section 2.2). We then present the process of protein palmitoylation, focusing on the regulation of its enzymatic machinery. Moreover, we discuss recent findings describing how the activity of (de-)palmitoylating enzymes is affected by altered metabolic states, such as during lipid-overload (Section 3.1, Section 3.2 and Section 3.3). Since protein palmitoylation within cardiac insulin signaling is virtually unexplored, we collect information from studies in (mainly) skeletal muscle and fat to summarize current knowledge over the physiological role of S-acylation within the insulin cascade and the uptake of energy substrates (Section 4). We also address the implication of aberrant protein palmitoylation in the onset of lipid-induced insulin resistance in skeletal muscle (Section 5). Lastly, in Section 6, we make use of the collected information to discuss the hypothetical role of S-acylation within cardiac insulin signaling and substrate uptake and whether protein palmitoylation may be an effective target to fight cardiac contractile dysfunction, which often concurs with lipid-induced insulin resistance.

## 2. Cardiac Insulin Signaling and Lipid-Induced Insulin Resistance

### 2.1. Cardiac Insulin Signaling

A simplified picture of the insulin cascade in the heart is described in this section and represented in Figure 1. Cardiac insulin signaling is initiated by the binding of insulin to the insulin receptor (IR), which subsequently interacts with and activates insulin receptor substrate 1 and 2 (IRS-1 and IRS-2) [16]. These first steps of the pathway are modulated by caveolins, which exist in three isoforms (caveolin-1, -2 and -3) and are attached via palmitoylation to plasma membrane subdomains called caveolae [17,18,19]. In the heart, the muscle-specific isoform caveolin-3 (Cav-3) interacts with IR and mediates cardiac downstream signaling starting with IRS-1 activation [20]. Subsequently, IRS-1/2 stimulate phosphatidylinositol-3 kinase (PI3K) to synthesize phosphatidylinositol-3,4,5-trisphosphate (PIP3) using phosphatidylinositol-4,5-bisphosphate (PIP2) as substrate [21]. The availability of PIP2 is dependent on the generation of its precursor phosphatidylinositol-4-phosphate (PI4P). PI4P is synthesized by the enzymes phosphatidylinositol 4-kinase II-alpha (PI4KIIα) and phosphatidylinositol 4-kinase II-beta (PI4KIIβ), which are both palmitoylated, thus allowing insertion in cellular membranes where PI4KIIα and PI4KIIβ produce PI4P [22,23]. PI4P is then converted into PIP2, which turns into PIP3 upon PI3K activation, hence triggering the phosphorylation and activation of Akt2 kinase [16]. In adipocytes, phosphorylation of Akt2 is enhanced by the interaction of the kinase with the palmitoylated protein Clip59-R at the plasma membrane [24]. Akt2 is a key kinase in the downstream regulation of the insulin cascade, and, once activated, Akt2 inhibits RabGAP AS160 allowing the de-inhibition of Rab proteins. In close proximity to the endosome, Rabs are essential for the formation of vesicles that move glucose and fatty acid transporters (GLUT4 and CD36, respectively) from their intracellular storage compartments to the plasma membrane. Once at the cell surface, vesicles fuse with the sarcolemma through vesicle-associated soluble *N*-ethylmaleimide-sensitive factor attachment protein receptors (v-SNAREs), which interact with specific target-membrane associated SNAREs (t-SNAREs), bringing vesicles in close proximity with the extracellular environment so that uptake of glucose, via GLUT4, and of fatty acids, via CD36, can occur [25].

Besides v- and t-SNAREs, the interaction between the plasma membrane and the substrate transporters appears to be influenced by palmitoylation of GLUT4 and of CD36 in adipocytes [26,27] and CD36 in the liver [28]. In the heart, there are specific vesicle-associated membrane protein (VAMP) isoforms dedicated to GLUT4 vesicle fusion, while other VAMPs are exclusively committed for the fatty acid transporter CD36. There are also VAMPs that mediate both GLUT4 and CD36 trafficking. For instance, VAMP2 is shared between cardiac GLUT4 and CD36 vesicles [29]. In adipocytes and rat brain, VAMP2 and its cognate t-SNAREs SNAP23 are attached to the respective vesicles and plasma membrane through palmitoylation [26,30]. The journey of GLUT4 and CD36 towards the sarcolemma appear to be similar. Still, there are subtle differences in their trafficking. Although endosomes represent the main intracellular storage for CD36, GLUT4 is additionally kept in multiple compartments named GLUT4 storage vesicles (GSVs), which either overlap with the endosomes or are found outside of this organelle [4,31]. In adipocytes, GSVs contain about half of the intracellularly stored GLUT4 (the other half in the endosome) [32]. Both endosomes and GSVs release GLUT4 in response to insulin, although the degree at which both GLUT4 and CD36 are expelled from the endosomes is further regulated by endosomal pH. Under normal conditions, the endosomal lumen is slightly acidic (pH ~5.5), which allows optimal endosomal retention of GLUT4 and CD36 and concomitant low rates of uptake of glucose and fatty acids under non-challenging conditions.

Additionally, properly acidified endosomes are optimally sensitive to increase reversible transporter translocation and substrate uptake in response to physiological stimuli such as insulin and increased contractile activity [33]. However, as further explained below, excess fatty acid supply to cardiomyocytes during lipid overconsumption causes loss of acidification of endosomes, initiating a series of events that culminate with the onset of cardiac-lipid-induced insulin resistance [34].

### 2.2. Cardiac Lipid-Induced Insulin Resistance

In response to a palmitic acid-enriched high fat diet, proper acidification of endosomes in cardiomyocytes is compromised. Consequently, CD36 is chronically expelled from the endosomes to redistribute to the plasma membrane [34]. The endosomal CD36 storage compartments that are consequently depleted appear to be the insulin-responsive compartments, resulting in high basal fatty acid uptake rates with a loss of insulin-stimulation of fatty acid uptake. Besides CD36, GLUT4 is also ejected from the endosomes under conditions of lipid overload. However, GLUT4 travels to other intracellular storage compartments, most likely GLUT4 storage vesicles (GSVs), without reaching the plasma membrane [31]. Increased trafficking of CD36 to the cell surface and the consequently chronically increased fatty acid uptake rates culminate into the intracellular accumulation of triacylglycerol, diacylglycerol (DAG) and ceramide (CER) of which the latter two lipid species inhibit the insulin cascade at the level of IRS-1 and Akt2 phosphorylation, respectively. Specifically, DAG accumulation activates IκB kinase (IKK) or c-Junk N-terminal kinase (JNK), both exerting inhibitory serine phosphorylation of IRS-1, thus hindering downstream insulin signaling. Intracellularly stored DAG inhibits IRS-1 also via the activation of the novel protein kinase-C (nPKC) isoform PKCθ [35]. In L6 myotubes, the insulin cascade is severely impaired by another member of the nPKC family, PKCε, which translocates to the nucleus and blunts the gene expression of IR in response to palmitate [36]. CER mainly acts on Akt2, impeding its translocation and activation to plasma membrane domains enriched in PIP3 [37]. As a result, downstream insulin signaling and insulin-stimulated RabGAP AS160 inhibition are impaired. Consequently, Rab proteins are not de-inhibited, rendering them to fail to initiate vesicle formation from GSVs, thereby imprisoning GLUT4 and blocking insulin-stimulated glucose uptake (Figure 2). This inhibition of downstream insulin signaling does not impact on CD36 localization since this transporter has already been kicked out of the endosomes. Thus, in lipid-overloaded cardiomyocytes, GLUT4 and CD36 display a juxtaposed localization, with the former being mainly stored intracellularly and the latter mainly present at the cell surface (Figure 2).

Some human studies indeed have confirmed the positive correlation between DAG/CER and insulin resistance, while other studies did not. For instance, *Vastus lateralis* DAG content in healthy subjects infused with lipids + insulin was higher compared with the group where the individuals were only administered with insulin [38]. Moreover, in response to the rise in DAG content, there was a significant increase in PKCs activity and a reduction in insulin sensitivity in the lipid infusion group compared with the non-infused subjects [38]. In another study, a cohort of obese people showed increased DAG muscle content and decreased insulin sensitivity compared with lean subjects [39]. A similar inversely proportional relationship was shown between CER and insulin sensitivity in muscle biopsies from lean and severely obese subjects [40]. Strikingly, inhibition of CER synthesis in mice fed a high-fat diet partially restored insulin-stimulated Akt phosphorylation in gastrocnemius muscles [41]. Despite this evidence, the correlation between CER/DAG and insulin resistance is not always explicit. For instance, analysis in rodents proved that DAG content and insulin sensitivity did not change in oxidative muscles between a mice model for insulin resistance and wild-type animals [42]. In humans, no difference in DAG concentration was found between muscle from insulin-resistant diabetics and muscle from non-diabetics [43]. In line with this, another human study demonstrated that there were no differences in DAG concentration between obese with impaired glucose tolerance and obese with normal glucose tolerance [44]. Most surprisingly, an even lower DAG content was found in obese people compared with the lean counterpart [45]. Taken together, these studies effectively separate DAG from insulin resistance.

Similarly, increased CER levels do not always correlate with insulin resistance. Infusion of soy oil to rats inhibited whole-body glucose uptake while they did not display any increase in CER content [46]. Moreover, among individuals with different insulin sensitivity, such as patients with type 2 diabetes, subjects with impaired glucose tolerance and healthy people, similar muscle CER contents were measured [47]. Another study detected higher CER levels in muscle from obese compared to lean individuals. However, after 30 or 45/60 min of insulin infusion, CER content leveled out between the two groups; still, the obese maintained lesser glucose infusion rates [48]. In general, there is conflicting evidence in the literature supporting a clear association between CER content and (muscular) insulin resistance [49] (Figure 2). The corollary of these observations combined is that alternative mechanisms, such as protein palmitoylation, should be taken into consideration as additional processes regulating insulin sensitivity during both healthy and altered metabolic states.

## 3. Palmitoylation and Its Enzymatic Regulation

Palmitoylation fulfills several biological functions and is finely regulated by enzymes that continuously attach and remove palmitate moieties to protein cysteines. Alteration in the activity of these enzymes results in dysregulation of palmitoylation of a specific protein(s) with physiological consequences.

### 3.1. Palmitoylation Function

A protein is defined as “palmitoylated” when a palmitic acid molecule is reversibly attached to a cysteine residue via a thioester bond. Biological effects resulting from this PTM modification vary from the regulation of protein trafficking between intracellular compartments to stabilization of integral membrane proteins and modulation of protein–protein interaction. The palmitate group is capable of inducing each of these effects thanks to its hydrophobicity causing membrane affinity. This particular property is fundamental in directing proteins to specific subcellular localizations, which determines their crucial involvement in regulating processes that are of pivotal importance for cellular function [50].

For instance, palmitoylation of endothelial nitric oxide synthase (eNOS) localizes this enzyme to the Golgi and plasma membrane, where it can produce and release nitric oxide, an important regulator of vascular homeostasis [51]. Furthermore, palmitoylation of tyrosine kinase Lck brings the enzyme to the plasma membrane, where it is essential for the activation of T cell receptor signaling in human T lymphocytes [52]. Moreover, protein palmitoylation governs several steps of the insulin cascade and consequently contributes to the regulation of recycling of substrate transporters between intracellular compartments and the plasma membrane.

Besides S-acylation, some proteins undergo other types of lipid modifications such as prenylation and *N*-myristoylation. However, unlike palmitoylation, these lipid modifications are irreversible modifications holding weaker hydrophobicity and membrane affinity compared to palmitoylation [14,53]. For this reason, protein prenylation and myristoylation alone cannot support stable membrane association of the protein, so that additional palmitoylation is commonly needed to provide proper membrane localization.

### 3.2. Palmitoylation Enzymatic Machinery

Palmitoylation is reversibly regulated by palmitoyl-acyltransferases (PATs) and acyl-protein thioesterases (APTs). PATs are the enzymes responsible for seizing palmitic acid from palmitoyl-CoA and transfer it to protein cysteines, while APTs mediate protein depalmitoylation through hydrolysis of the acyl thioester. There are 23 known mammalian PATs having in common a highly conserved catalytic Asp-His-His-Cys (DHHC) cysteine-rich domain [54] and, for this reason, are named DHHC1 to DHHC24 (DHHC10 missing) [10]. Each of the 23 DHHCs retains a specific subcellular localization: these enzymes are found in the endoplasmic reticulum (ER), Golgi apparatus, endosomes and plasma membrane, with Golgi being the major hub for protein palmitoylation [55]. On the other hand, APTs are present in much lower numbers, and their localization is less well defined when compared to PATs. The most studied depalmitoylating enzymes are acyl-protein thioesterase 1 (APT1) [56] and acyl-protein thioesterase 2 (APT2) [57], which are mostly scattered throughout the cytosol, but partially localize to the Golgi and are responsible for the majority of deacylation events within cells [58]. Palmitoyl protein thioesterase 1 (PPT1) is found almost exclusively in late endosome/lysosome and is known to depalmitoylate only a small number of proteins [59,60]. Contrary to APTs, PATs display a higher level of substrate specificity, in line with their family member abundance and partly due to their defined subcellular localization. Each DHHC mediates palmitoylation of a defined set of protein target(s), which in turn can be palmitoylated by more than one DHHC. Conversely, given that only two thioesterases (APT1 and APT2) are responsible for intracellular depalmitoylation processes and that these enzymes also are more randomly distributed throughout the cell, it seems unlikely that APTs exhibit as much specificity as PATs.

With respect to the heart, there is relatively little information on abundance, localization and substrate specificity of the different PAT members, but nonetheless, several observations have been made. For instance, DHHC5 in the heart is located at the plasma membrane where it palmitoylates and thus influences the activity of the protein phospholemman (PLM), a regulator of the sarcolemmal Na^+^ pump [61]. Additionally, DHHC2 contributes to PLM regulation [62]. DHHC5 also mediates palmitoylation of Gα proteins at the sarcolemma in response to β-adrenergic stimulation, thus influencing downstream signaling [63]. Although less abundant in the heart [61], the ER-resident DHHC16 palmitoylates phospholamban (PLN) in the sarcoplasmic reticulum (SR). S-acylation allows PLN to interact with and regulate the activity of the Ca^2+^-ATPase SERCA2a [64]. As a result, the known involvement of palmitoylation in the heart appears to involve aspects of cardiac electrophysiology and β-adrenergic signaling [65], whereas the role of protein S-acylation within cardiac insulin signaling remains rather unexplored. In contrast, protein palmitoylation modulates the insulin signaling cascade in skeletal muscle and adipocytes (see Section 4). In next Section 3.3, the enzyme activity regulation of PATs and APTs will be discussed so as to explore how possible defects in these enzyme activities may compromise protein palmitoylation patterns.

### 3.3. Regulation of PATs and APTs

Prior to mediating palmitoylation of specific protein targets, PATs undergo autopalmitoylation, which appears a critical event [66]. The majority of PATs have four transmembrane (TM) domains, with the catalytic DHHC domain facing the lumen of vesicular compartments [67,68]. It is believed that the enzyme performs the palmitoylation reaction via a so-called ping-pong mechanism [66]. First, PATs attack palmitoyl-CoA, thus allowing autoacylation of the enzyme at cysteine residues within the DHHC domain of the enzyme and forming the intermediate PAT-palmitate. Subsequently, palmitate is transferred to the cysteine of the protein substrate via a thioester bond [69,70] (Figure 3). However, not all PATs undergo the autopalmitoylation process to form the PAT-palmitate intermediate. For instance, studies performed on the two PATs DHHC13 and DHHC17, failed to detect any palmitoylation of those isoforms within the DHHC domain [71,72].

Palmitoylation of PATs takes place at cysteine residues outside of the DHHC region [73]. For instance, in the heart, palmitoylation of DHHC5 occurs within its long C-terminal tail and stabilizes the enzyme at the plasma membrane, thereby influencing its activity-dependent localization for the regulation of β-adrenergic signaling [63]. In addition, palmitoylation of hippocampal DHHC3 increased the activity of the enzyme towards its protein substrate AMPA glutamate receptor subunit GluA1. This latter study did not investigate which region of DHHC3 was palmitoylated [74], although it is known that mutation of cysteines outside the DHHC motif results in activity deficit and structural perturbation of DHHC3 in Sf9 insect cells [75]. Therefore, palmitoylation outside of the DHHC catalytic domain might be the potential mechanism enhancing PATs activity and/or specificity [73] (Figure 3). Contrary to PATs, palmitoylation has a negative influence on the activity of selected acyl protein thioesterases (APTs) in removing palmitate from cysteines. Specifically, APTs are compartmentalized into two intracellular pools located in the cytosol and in the Golgi, respectively, of which the former is unpalmitoylated and able to deacylate peripheral membrane proteins. Instead, the APTs localized at the Golgi are palmitoylated and inactive towards other protein substrates [58]. Accordingly, the endosomal/lysosomal thioesterase PPT1 increased its thioesterase activity when unpalmitoylated [76]. Furthermore, recent insights revealed a higher level of complexity of the regulation of PATs and APTs. Namely, the existence of a “palmitoylation cascade” has been postulated based on observations in HeLa cells [77]. Specifically, DHHC16 mediates the palmitoylation of DHHC6 at three different cysteine sites, thus leading to the formation of various DHHC6 forms, each one having a specific activity [77]. Collectively, these observations suggest that palmitate itself (i.e., substrate availability) plays an important role in regulating PATs and APTs (Figure 3).

Importantly, alteration in PATs/APTs activity can lead to dysregulation of protein palmitoylation, ultimately having an impact on protein function. Recently, it was shown that DHHC3 autopalmitoylation could be regulated by phosphorylation. Both from studies using N2a cells and from observations in in vivo mouse brains, it was found that mutagenesis of two DHHC3 phosphorylation sites increased both DHHC3 autopalmitoylation and enzymatic activity and augmented NCAM S-acylation, the protein target of DHHC3. This resulted in the stimulation of neurite outgrowth [78]. DHHC3 (auto)palmitoylation and activity are also dysregulated in response to increased palmitate availability. High-fat diet (HFD)-induced insulin resistance favored palmitate accumulation in the mouse brain, thus enhancing DHHC3 palmitoylation and activity, which, in turn, caused hyper-palmitoylation of AMPA glutamate receptor subunit GluA1. This led to an impairment of GluA1 trafficking to and functioning at the plasma membrane, finally hindering hippocampal synaptic plasticity [74]. In *db/db* mice, a model of type 2 diabetes showing marked insulin resistance, APT1 thioesterase activity is impaired, resulting in increased R-Ras palmitoylation and retention at the plasma membrane [79]. These latter two examples [74,79] showed that activity of the enzymatic palmitoylation machinery is susceptible to feedforward regulation, thereby making protein palmitoylation an extremely sensitive indicator of lipid excess, as occurs in insulin resistance. Importantly, as elaborated in Section 2.2, (cardiac) insulin signaling is markedly influenced by substrate availability, i.e., inhibited by an excess of circulating lipids.

In Section 4, we will provide an overview of palmitoylation of individual proteins within the insulin signaling cascade and, in the subsequent vesicular trafficking, the process underlying insulin-stimulated substrate transporter translocation under low lipid exposure (i.e., normal healthy condition). Protein palmitoylation under low lipid exposure will be referred to as basal protein palmitoylation. Additionally, both substrate transporters GLUT4 and CD36 undergo basal palmitoylation with consequences for their translocation, which also will be described in this section. In Section 5, several examples of aberrant/excess protein palmitoylation are illustrated, and their role in the onset of insulin resistance is discussed.

## 4. Role of Palmitoylation of Signaling and Trafficking Proteins in Insulin-Stimulated Substrate Transporter Translocation

This section provides a list of proteins in the insulin signaling pathway and subsequent trafficking process towards translocation of GLUT4 and CD36, which undergo basal palmitoylation. Additionally, palmitoylation of both transporters themselves is discussed. Per the listed protein, it is detailed which PATs and APTs are involved and how this modulates its functioning.

### 4.1. Caveolins

As outlined in Section 2.1, caveolins are a group of proteins localized in specific plasma membrane subdomains called caveolae [80,81]. These membrane invaginations are important hubs for the formation of signaling complexes that initiate intracellular cascades in response to growth factors and hormones such as EGF [82], PDGF, and insulin [17,83]. Caveolins (Cav-1, Cav-2 and Cav-3) share a common structure in which the C-terminal domain undergoes palmitoylation at specific cysteine residues [19,84]. Palmitoylation provides caveolins with proper localization within the plasma membrane. Various studies suggest that these proteins contribute to modulating the insulin cascade.

*Caveolin-1:* Cav-1 is palmitoylated at different cysteine residues [85] by DHHC7 and DHHC21 [86] and is considered a representative marker for palmitoylated proteins [87,88,89]. Palmitoylation is essential for Cav-1 both to be stabilized at the plasma membrane [18] and to further interact with c-Src kinase [90], which phosphorylates Cav-1 as observed both in vivo [91] and in 3T3-L1 adipocytes [92]. In skeletal muscle, insulin-stimulated c-Src kinase activates PKCδ kinase, which then interacts with IR to trigger its phosphorylation to induce glucose uptake [93,94]. Curiously, in fibroblast, Cav-1 phosphorylation (triggered by the presence of palmitate in adipocytes [92]) is the signal for the recruitment of C-terminal Src kinase (Csk), which inhibits c-Src activity [95]. Thus, based on the collected information, we sketched a hypothetical scheme in which palmitoylation-dependent phosphorylation of Cav-1 may play a role in activating negative feedback (through the inhibition of c-Src kinase activity [95]) aiming to mitigate excessive insulin-stimulated glucose uptake (Figure 4).

*Caveolin-2:* Cav-2 is palmitoylated at three different cysteines in fibroblast and adipocytes, but, contrary to Cav-1, the responsible DHHCs are currently unknown. After synthesis in the ER, Cav-2 also undergoes myristoylation. The latter is a necessary precondition for Cav-2 to be palmitoylated at the Golgi and subsequently translocate to the plasma membrane where it is phosphorylated by IR. The latter process promotes IRS-1 interaction with both Cav-2 and IR, facilitating the phosphorylation and activation of IRS-1 by IR. Palmitoylation-dependent Cav-2 phosphorylation, therefore, modulates IRS-1 interaction with and activation by IR [96]. Cav-2 is expressed in fibroblasts, adipocytes, and endothelial cells [97]. In the heart, Cav-2 is present mostly in the vascular endothelium [98] but also found in cardiomyocytes [99]. However, palmitoylation of cardiac Cav-2 and its putative role in modulating the insulin cascade remain unexplored. Cav-3 is a muscle-specific caveolin isoform highly expressed in cardiomyocytes [97]. Similarly, as Cav-2, Cav-3 is palmitoylated at the plasma membrane [100]. Yet, no DHHCs are associated with modification of Cav-3. Moreover, the role of Cav-3 palmitoylation in insulin signaling has not been studied yet.

### 4.2. Phosphatidylinositol 4-Kinase II-α

Phosphatidylinositol 4-kinase II-alpha (PI4KIIα) catalyzes the production of PI(4,5)P_2_ from PI4P. PI(4,5)P_2_ generation is crucial for proper insulin signaling as this phospholipid species is subsequently converted by PI3K into PIP3, which is needed for plasma membrane-binding and activation of Akt [4,101]. PI4KIIα localizes and acts on the Golgi thanks to the palmitoylation of its cysteine-rich region, CCPCC, located on the protein catalytic subunit [102]. This process is controlled by two Golgi-localized PAT enzymes, DHHC3 and DHHC7. Overexpression of DHHC7 increased the palmitoylation of PI4KIIα [22]. Palmitoylation enhanced the enzymatic activity of PI4KIIα by increasing the stability of the catalytic region where *phosphatidylinositol* (PI) binds and is converted to PI4P [102]. Inhibition of palmitoylation by 2-bromopalmitate accordingly decreased PI4P generation by PI4KIIα [103]. In conclusion, PI4KIIα palmitoylation may positively contribute to enhancing insulin signaling (Figure 5).

### 4.3. ClipR-59

ClipR-59 is a cytoplasmic linker protein characterized by its two cytoskeleton-associated protein glycine-rich (CAPGly) domains and a membrane-binding domain (MBD) at the C-terminal. The protein is dually palmitoylated at Cys 534 and Cys 535 located at the C-terminal of the protein [104]. COS-7 and HEK293 cell culture studies revealed two potential PATs to modulate palmitoylation of Clip59-R, i.e., DHHC13 and DHHC17, with the latter to display higher affinity making it the major contributor to ClipR-59 palmitoylation [105]. Palmitoylation allows ClipR-59 to be localized at the plasma membrane where it interacts with Akt promoting its phosphorylation upon insulin stimulation (Figure 5) [24]. In adipocytes, inhibition of Clip59-R palmitoylation via DHHC17 silencing caused a marked decrease in plasma membrane content of Clip59-R and Akt phosphorylation, finally hindering GLUT4 translocation to the cell surface [105]. Overall, Clip59-R palmitoylation has an important function as a stimulating factor for insulin signaling.

### 4.4. Tumor Suppressor SCRIB and Phosphatase SCP1

In endothelial cells, the tumor suppressor protein SCRIB and the phosphatase SCP1 undergo palmitoylation [106,107]. SCRIB is palmitoylated by DHHC7, favoring its plasma membrane localization and subsequent inhibition of PI3K/Akt signaling (Figure 5) in response to epidermal growth factor (EGF) [106]. SCP1 phosphatase is palmitoylated by a yet unknown DHHC [107]. Palmitoylated SCP1 interacts with phospho-Akt at the plasma membrane leading to de-phosphorylation and de-activation of this kinase during insulin stimulation (Figure 5). This further results in impaired angiogenesis and tumorigenesis. In line with this, unpalmitoylated SCP1 mutants lose their capacity to inhibit the phosphorylation of Akt [107]. Finally, in HEK293T and H1299 cells, treatment with the palmitoylation inhibitor 2-bromopalmitate augmented insulin-stimulated Akt phosphorylation [107], but it has not been assessed whether this is due to the reversal of the inhibitory effects of SCRIB or SCP1 palmitoylation on Akt phosphorylation.

### 4.5. SNARE Proteins

As described in Section 2.1, GLUT4 subcellular vesicles (GSV) in cardiomyocytes contain VAMP-2, which interacts with its cognate target membrane-associated SNAREs (t-SNAREs) SNAP23 and syntaxtin4 to promote the tethering of GLUT4 with the sarcolemma [31]. VAMP-2, SNAP23 and syntaxtin4 each are palmitoylated in adipocytes [26,108]. SNAP23 S-acylation was also confirmed in HeLa cells [109], HEK cells [110] and PC12 cells [111]. Inhibition of palmitoylation abrogated VAMP-2 plasma membrane association and decreased glucose uptake in adipocytes [108], whereas in neuronal cells, deficiency of PPT1 thioesterase resulted in persistent VAMP2 palmitoylation and synaptic vesicle retention at the plasma membrane, hindering SV recycling and leading to infantile neuronal ceroid lipofuscinoses [30,112]. In 3T3-L1 adipocytes, SNAP23 overexpression increased both insulin-stimulated GLUT4 content at the plasma membrane and glucose uptake [113]. In that latter study, it was hypothesized that, because SNAP23 is stabilized at the plasma membrane via palmitoylation [110], this PTM may play an important role in the observed increase in plasma membrane GLUT4 as well as in glucose uptake [113].

As summarized above, the role of v- and t- SNARE palmitoylation in modulating GLUT4 trafficking and glucose uptake has been so far only explored in fat cells. However, adipocytes [108] and HEK cells [110] provide information concerning the PATs associated with VAMP2 and SNAP23, which are similarly expressed in the heart [114,115]. In adipocytes, VAMP2 is palmitoylated by the Golgi-localized DHHC7 (Figure 5) [108]. DHHC7 also palmitoylates SNAP23 at the Golgi, together with DHHC3 and DHHC17, providing this t-SNARE with the necessary anchor for plasma membrane localization. At the cell surface, SNAP23 becomes stabilized by DHHC2 (Figure 5) [110]. The v- and t- SNARE palmitoylation appears to be complex processes, showing redundancy at the level of DHHC7 (shared between VAMP2 and SNAP23), thus making this enzyme a potential target to modulate the function of proteins involved in GLUT4 and CD36 trafficking.

### 4.6. GLUT4

GLUT4 is the main glucose transporter in adipose tissue [116], skeletal muscle [117,118], and heart [119]. Insulin-stimulated translocation of GLUT4 from GSV to the plasma membrane is modulated by Rab proteins, which are activated by the inhibition of AS160, as detailed in Section 2.1. However, GLUT4 trafficking appears to have a more complex regulation. Palmitoylation of GLUT4 has been described to be an additional and essential physiological factor for the translocation of the protein from cytosolic vesicles to the plasma membrane. Specifically, in 3T3-L1 and primary adipocytes, GLUT4 is palmitoylated at Cys223 by DHHC7 (Figure 5). Inhibition of palmitoylation by either C223S substitution or DHHC7 silencing resulted in a significant reduction of GLUT4 at the plasma membrane upon insulin stimulation and, consequently, diminished glucose uptake [108,120].

In addition to GLUT4, insulin stimulates glucose uptake via another glucose transporter, GLUT1, but supposedly not by regulation of its plasma membrane localization [121,122]. Interestingly, GLUT1 is palmitoylated in human peripheral blood mononuclear cells [123] and blood–brain barrier capillaries [124]. GLUT1 palmitoylation may be involved in GLUT1-mediated glucose uptake but has not yet been studied in adipocytes and cardiomyocytes.

### 4.7. CD36

The fatty acid transporter CD36 is well-known to be palmitoylated at its N-terminal (Cys 3, 7) and C-terminal (Cys 464, 466) domains [125]. Palmitoylation is essential for CD36 processing and maturation in the endoplasmic reticulum and further trafficking to the Golgi [126]. Moreover, S-acylation allows CD36 to be translocated from the Golgi to the plasma membrane. Specifically, in adipocytes, DHHC4 palmitoylates CD36 in the Golgi apparatus to induce CD36 translocation to the plasma membrane, whereas DHHC5 palmitoylation is required to stabilize CD36 at the plasma membrane, thus commencing the uptake of fatty acids (Figure 5) [127]. Upon increased fatty acid supply to adipocytes and their binding to CD36, DHHC5 becomes phosphorylated and inactivated, leading to depalmitoylation of CD36 by APT1-thioesterase. This initiates the internalization of fatty acid-loaded CD36 by endocytosis. Besides the fatty acids directly taken up via CD36 at the cell surface [128], these endocytotically taken-up fatty acids via CD36 also contribute to cellular lipid accumulation in adipocytes. Hence, palmitoylation/depalmitoylation cycles of CD36 finely tune the process of fatty acid uptake [129].

Yet, a study with CD36-Cys mutants in CHO cells revealed that although all four terminal Cys residues within CD36 are essential for insulin-stimulated CD36 translocation, palmitoylation of these Cys residues was not involved [130]. Perhaps, the involvement of CD36 palmitoylation in insulin-stimulated CD36 translocation is cell-type specific. Further studies are needed to reveal the role of CD36 palmitoylation in the heart.

### 4.8. Corollary for Insulin-Stimulated Substrate Transporter Translocation

In general, most of the above described basal palmitoylation events positively contribute to insulin signaling regulation and insulin-induced substrate transporter translocation. Yet, there are also examples of listed proteins with opposing effects on insulin signaling. For instance, Cav-1 palmitoylation may exert a possible negative effect on glucose uptake, while Cav-2 palmitoylation positively contributes to correct the unwinding of the insulin cascade (Section 4.1). Whether these opposing effects occur simultaneously needs to be further investigated.

Another example is given by the regulation of Akt activity by either Clip59-R or SCRIB/SCP1. Clip59-R palmitoylation by DHHC17 favors Akt phosphorylation (Section 4.3). In contrast, either palmitoylation of SCP1 by a yet unknown DHHC or palmitoylation of SCRIB by DHHC7 abolishes Akt phosphorylation (Section 4.4). This elicits a scenario whereby DHHC17 positively impacts the insulin signaling pathway (via enhanced Clip59-R palmitoylation), whereas DHHC7 appears to negatively regulate the cascade (via Cav-1 palmitoylation-s and SCRIB palmitoylation). Conversely, within the context of insulin-induced substrate transporter translocation, DHHC7 positively contributes to the mechanisms of GLUT4 trafficking (Section 4.6) and GLUT4 vesicles docking and fusion with the plasma membrane (Section 4.5). In addition to GLUT4, DHHC7 is also important for the plasma membrane translocation of another major cargo protein for GLUT4 vesicles, named insulin-responsive aminopeptidase (IRAP) [108], which was found to be palmitoylated in adipocytes [26]. Hence, DHHC17 is a positive stimulus of insulin signaling, while DHHC7 appears as a negative regulator. On the other hand, DHHC7 positively regulates GLUT4 trafficking, thus making DHHC7 an interesting palmitoylating enzyme with opposite effects on insulin signaling (inhibitory) and GLUT4 plasma membrane translocation (stimulatory).

Finally, other components of the signaling and trafficking pathways towards insulin-stimulated transporter translocation have been found to be palmitoylated, such as AS160 (Section 2.1) and MUNC18 (an adaptor for SNARE-mediated fusion of GLUT4 vesicles with the plasma membrane), but the roles of these palmitoylations in insulin signaling have not yet been investigated [26,131]. Overall, basal palmitoylation appears to positively contribute to insulin signaling and subsequent substrate uptake, and the list of palmitoylated proteins (as displayed in Table 1) in these processes is likely to expand with future investigations.

## 5. Aberrant Protein Palmitoylation Driving Insulin Resistance

The conflicting pieces of evidence concerning DAG and CER as the sole factors driving the onset of lipid-induced insulin resistance (discussed in Section 2.2) leave room to investigate other maladaptive mechanisms involved in the inhibition of the insulin cascade. During lipid overload, there will be aberrant/excessive palmitoylation of proteins, given that the palmitoylation machinery operates in a substrate-driven manner (Section 3.3). Below is a list of examples of proteins for which excessive palmitoylation would negatively contribute to insulin signaling. This list (Table 2) also includes both substrate transporters.

### 5.1. PKCε

PKCε is a member of the group of novel PKC kinases (nPKCs) [134]. This nPKC is particularly relevant in cardiac muscle since sarcomeric elements such as cardiac troponin T and I (cTnT and cTnI), desmin, and myosin light chain-2 are found in complex with PKCε [135]. Moreover, PKCε-mediated Ser-43/Ser-45 phosphorylation in cTnI is critical for normal Ca^2+^ sensitivity of myofilaments [136]. Besides this role in cardiac muscle contraction mechanisms, PKCε is known to be involved in the onset of insulin resistance in response to chronic diacylglycerol (DAG) accumulation. In skeletal muscle of diabetic animals [137] and palmitate-overloaded adipocytes [138], an intracellular increase in DAG content leads to an over-activation of PKCε. This, in turn, caused downregulation of IR expression, which decreased insulin sensitivity [137,139].

Further investigations on the molecular mechanisms involved have shed light on the involvement of palmitoylation as a key mediator of PKCε-mediated IR expression inhibition. Specifically, PKCε phosphorylation was increased in palmitate-overloaded L6 myotubes, and this aberrant phosphorylation was caused by the dual palmitoylation of PKCε. Phospho-PKCε is then transported to the nuclear region, where it diminishes IR gene expression, thus compromising insulin-stimulated GLUT4 translocation and glucose uptake. Hence, palmitoylation-dependent PKCε phosphorylation is a key regulator for the development of insulin resistance in skeletal muscle [36].

### 5.2. GAPDH

Glyceraldehyde-3-phosphate dehydrogenase (GAPDH) governs the sixth step of glycolysis [140]. A study with purified GAPDH in cell-free systems showed that increasing palmitoyl-CoA concentrations enhanced GAPDH palmitoylation at Cys-244, thereby targeting it to membranes and inhibiting its enzymatic activity [133]. Therefore, GAPDH inhibition via palmitoylation could be taken into consideration as an additional mechanism through which accumulation of acyl(palmitoyl)-CoA during lipotoxicity leads to decreased glucose utilization and the onset of insulin resistance [133].

### 5.3. GLUT4

As mentioned in Section 4.6, basal palmitoylation of GLUT4 positively contributes to insulin-stimulated GLUT4 translocation. In adipocytes from obese mice fed with a high-fat diet, GLUT4 palmitoylation is increased [26]. In the context that excess lipids impair GLUT4 translocation in fat cells [141,142] and in cardiomyocytes [143], this hyper-palmitoylation of GLUT4 would exert a negative action on insulin-stimulated GLUT4 translocation. The link between GLUT4 hyper-palmitoylation and hindered trafficking needs to be investigated further.

### 5.4. CD36

While the role of basal CD36 palmitoylation in the regulation of insulin signaling is not unequivocal yet (Section 4.7), there is strong evidence that CD36 hyper-palmitoylation negatively impacts insulin sensitivity. In livers of patients with non-alcoholic steatohepatitis (NASH) and in mouse NASH models, CD36 was excessively palmitoylated and increasingly localized at the plasma membrane of hepatocytes, leading to excessive fatty acid uptake. Inhibition of CD36 palmitoylation prevented excessive fatty acid uptake and the development of NASH [28]. In parallel, excessive CD36 palmitoylation strengthened the interaction between CD36 and the tyrosine kinases Fyn and Lyn at the plasma membrane. The formation of this ternary complex inhibited the activation of AMPK, resulting in a reduction of fatty acid β-oxidation in the liver [28]. Hence, the combination of increased fatty acid uptake with decreased fatty acid oxidation upon CD36 hyper-palmitoylation inevitably results in massive steatosis in the liver, contributing to the development of NASH.

### 5.5. Corollary for the Role of Palmitoylation in the Development of Insulin Resistance

Taking Section 4 and Section 5 together, here is an apparent paradox on the role of protein palmitoylation in (cardiac) cellular insulin sensitivity, being that a basal degree of palmitoylation appears stimulatory, while excessive palmitoylation appears inhibitory. As the first attempt to resolve this paradox, the opposite effects of basal and excessive palmitoylation of CD36 may be considered. Under basal low lipid conditions, ~50% of the total CD36 population is present at the cell surface, which is necessary for basal energy provision of the heart. Basal CD36 palmitoylation appears essential for contributing to such subcellular distribution. A further increase in CD36 palmitoylation leads to increased CD36 cell surface localization and concomitantly increased fatty acid uptake, at least in the liver [28]. CD36 accumulation at the plasma membrane typically represents the first step leading to the onset of lipid-induced insulin resistance partly via CER or DAG (see Section 2.2).

As mentioned, basal palmitoylation of most of the listed proteins in Section 4 appears necessary for optimal functioning of specific components of the insulin signaling and trafficking pathways towards transporter translocation, but it is not known whether further increased levels of palmitoylation of these proteins will alter (i.e., decrease) their function, and thereby resolve the palmitoylation paradox in insulin sensitivity.

Another possibility to resolve this paradox is a scenario that several signaling/trafficking proteins, while not palmitoylated during basal conditions, become palmitoylated only during lipid overload. This may relate to weaker palmitoylation consensus motifs within these proteins, rendering them less accessible substrates for PATs. Subsequently, palmitoylation of these proteins during excess lipid supply may then negatively impact insulin signaling. PKCε, as a negative regulator of insulin signaling (see Section 5.1), may provide a possible example.

## 6. Conclusions and Future Perspectives

As revealed in studies in adipocytes and skeletal muscle, many proteins of the insulin signaling cascade and trafficking machinery involved in the translocation of GLUT4 and CD36 to the plasma membrane appear to be modulated by palmitoylation. However, in the heart, the role of S-acylation within the insulin signaling pathway remains relatively unexplored. Remarkably, the same PATs responsible for protein palmitoylation within the insulin cascade in adipose tissue and skeletal muscle are also among the most abundant PATs in the heart. For instance, palmitoylation of CD36 in adipocytes is mediated by DHHC4 and DHHC5 [127], which are, respectively, the second and the third most abundantly expressed PATs in rat cardiomyocytes [61]. Moreover, one of the PATs dedicated to SNAP23 palmitoylation is DHHC2 [110], which also is the most abundant PAT in cardiomyocytes [61]. GLUT4, caveolin-1, VAMP2 and PI4KIIα are palmitoylated by DHHC7 [22,86,108], which is also highly expressed in rat cardiac muscle [61]. Although not the most prominent isoform in the heart, DHHC17 [61] is a PAT of potential interest because, together with DHHC7 (Section 4.3 and ref. [106]), DHHC17 is involved in the regulation of Akt phosphorylation, as observed in adipocytes (Section 4.4 and ref. [105]). Hence, future work is required to investigate the role of palmitoylation in the context of cardiac insulin signaling and substrate uptake.

Given that the palmitoylation machinery is a sensitive sensor of extracellular lipid supply and likely a contributing factor to lipid-induced insulin resistance, this machinery might therefore provide novel attractive targets for combatting this maladaptive metabolic state. In particular, the reversible nature of the palmitoylation process might offer advantages in the diagnosis and treatment of lipid-induced insulin resistance in the heart and other tissues. Since there are at least 23 PAT isoforms, each one with a definite subcellular localization and specific protein(s) target(s) to palmitoylate, the possibility exists of targeting specific PATs. In this way, several disease states in which aberrant palmitoylation is a contributing factor can be treated in a more specific manner. For instance, DHHC3 modulation could be an effective strategy to prevent the aberrant palmitoylation of integrin (ITG) α6 and β4, which contributes to cancer development [144]. Furthermore, DHHC17 is considered an oncogenic PAT which could be targeted to impede aberrant palmitoylation of Ras [145]. On the other hand, targeted modulation of thioesterases (APTs) to restore aberrant protein palmitoylation seems to be less suitable compared to targeting PATs due to the non-specificity of APTs (non-specificity of APTs discussed in Section 3.2). With respect to lipid-induced insulin resistance, DHHC4 and DHHC5 may be pharmacologically inhibited to reduce CD36 palmitoylation and associated CD36 translocation. This would lead to decreased fatty acid uptake and myocellular lipid accumulation. Furthermore, the concomitant inhibition of DHHC7 and stimulation of DHHC17, may aid in restoring blunted Akt phosphorylation during lipid-overload condition (DHHC7-inhibitory and DHHC17-stimulatory effects on Akt phosphorylation are discussed in Section 4.8). Restoration of Akt activity via PATs would improve (insulin-stimulated) glucose uptake. Subsequently, improved glucose influx into cardiomyocytes would lead to re-acidification of the endosomes (as explained in. [142]), leading to reinternalization of CD36 (see Section 2.1) and decreased fatty acid uptake. This would be beneficial for the lipid-overloaded (pre-)diabetic heart. Notwithstanding, targeting of PATs should be carefully planned as restorative approach because, although more specific than APTs, some PATs are capable of palmitoylating multiple proteins. For instance, as discussed in the Section 4.2 and Section 4.8, DHHC7 palmitoylates four targets, (SCRIB, Cav-1, PI4KIIα and GLUT4) producing opposite effects within insulin signaling and substrate transporter trafficking.

Despite these limitations, still, specific PATs may be attractive targets to improve insulin resistance and normalize substrate transport in the heart. However, pharmacological inhibitors/activators of specific DHHC isoforms are not available yet. Just recently, innovative studies on PATs structure have been undertaken. These studies may speed up the realization of targeted-DHHCs drugs [146]. Selective control of DHHC activity may represent an innovative strategy to modulate insulin signaling and to fight lipid-induced cardiac contractile dysfunction in obese and diabetic individuals.

## Figures and Tables

**Figure 1 ijms-21-09438-f001:**
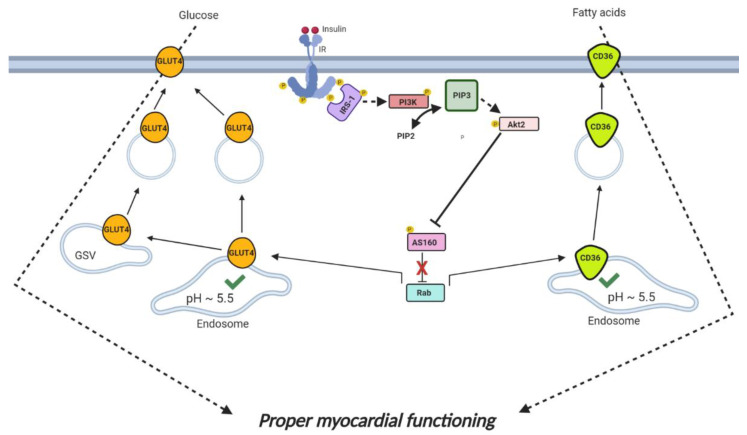
Cardiac insulin signaling pathway and substrate uptake. Binding of insulin to the insulin receptor (IR) stimulates IR autophosphorylation, which activates insulin receptor substrate-1 (IRS-1). Phosphorylated IRS-1 then stimulates phosphatidylinositol 3-kinase (PI3K) to convert the lipid metabolite PIP2 into PIP3, the activator of Akt2 kinase. Once activated, Akt2 performs inhibiting phosphorylation on RabGAP AS160, thereby de-inhibiting Rab proteins, which trigger the budding of vesicles containing GLUT4 (in orange) and CD36 (in light green) substrate transporters from the endosome. Budding of GLUT4 vesicles in response to insulin also happens from an additional GLUT4 intracellular storage compartment, namely GLUT4 storage vesicles (GSVs). Following vesicle formation, GLUT4 and CD36 will travel to the sarcolemma to mediate the uptake of glucose and (long-chain) fatty acids, respectively. The fusion between GLUT4- or CD36-containing vesicles and the sarcolemma is mediated by v- and t-SNARE proteins (not represented in the figure). The tuned subcellular recycling of these substrate transporter molecules to and from the sarcolemma translates into balanced glucose and fatty acid uptake rates, which is necessary for proper myocardial functioning.

**Figure 2 ijms-21-09438-f002:**
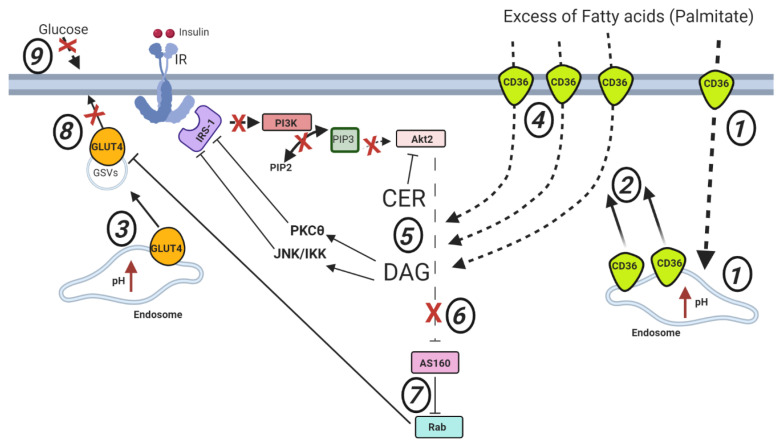
Putative model for the development of (cardiac) lipid-induced insulin resistance. Increasing concentrations of extracellular fatty acids (mostly palmitate) stimulate the uptake of lipids into myocytes via CD36, resulting in an increase in endosomal pH (1). This loss of endosomal acidification (*red arrow*) causes loss of endosomal CD36 retention, thus allowing an abnormal amount of CD36 molecules to relocate to the sarcolemma (2). Loss of endosomal acidification also impacts GLUT4, which is kicked out from the endosome to relocate to GSVs (3). The increased presence of CD36 at the plasma membrane (4) causes an increase in fatty acid uptake and subsequent accumulation of lipid species such as diacylglycerol (DAG) and ceramide (CER). DAG is known to inhibit IRS-1 via activation of JNK/IKK and PKCθ, finally hindering the insulin signaling pathway. CER acts on Akt2 (5), making the kinase unable to exert the inhibitory phosphorylation on RabGAP AS160 (6). Consequently, Rab protein activity is impaired (7), thus halting the translocation of GLUT4 vesicles from GSVs to the sarcolemma (8). Accordingly, glucose uptake is inhibited (9). Note the resultant juxtaposed localization of CD36 and GLUT4 in the insulin-resistant myocyte.

**Figure 3 ijms-21-09438-f003:**
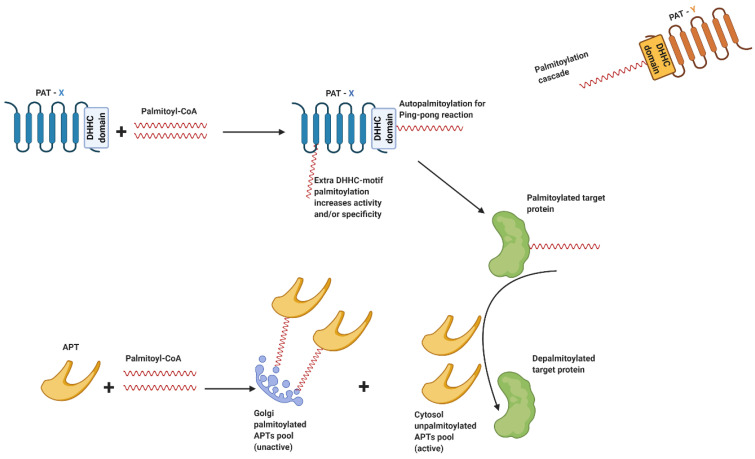
Model of regulation of PATs and APTs. *Upper part*: palmitoyl acyltransferase-X (PAT-X in blue) attacks palmitoyl-CoA and performs autopalmitoylation within the DHHC domain to form the acyl-intermediate of the so-called ping-pong reaction. Palmitate is then transferred to protein cysteines. Besides palmitoylation within the DHHC motif, S-acylation of PAT can also occur in other regions of the enzyme (extra DHHC-motif palmitoylation), presumably as a mechanism to increase PATs activity and/or specificity. In addition, (auto)palmitoylation of PATs may be mediated by other PATs (PAT-Y in orange) in a reaction identified as palmitoylation cascade. *Bottom part*: When palmitoylated, acyl protein thioesterase (APT in light orange) is anchored to the Golgi, forming the Golgi unactive APT pool. In contrast, the unpalmitoylated cytoplasmic pool of APTs is active and performs depalmitoylation of target proteins.

**Figure 4 ijms-21-09438-f004:**
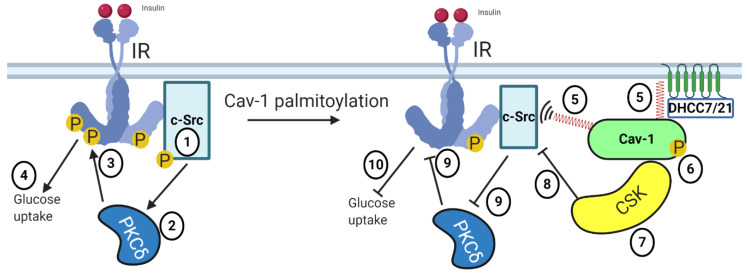
Hypothetical role of Cav-1 palmitoylation within the insulin signaling cascade. In the absence of Cav-1 palmitoylation, insulin stimulates c-Src kinase phosphorylation (1), triggering PKCδ kinase activation (2). Both c-Src and PKCδ kinases contribute to the stimulation of IR phosphorylation (3), ultimately inducing glucose uptake (4). Palmitoylation of Cav-1 (by DHHC7/21) allows interaction with c-Src (5), which then phosphorylates Cav-1 (6). Cav-1 phosphorylation is the signal for the recruitment of C-terminal Src kinase (Csk) (7), which inhibits c-Src kinase (8). Consequently, activation of PKCδ and IR phosphorylation are hindered (9), which mitigates insulin-stimulated glucose uptake (10). (Palmitate moieties represented as spring-red shape).

**Figure 5 ijms-21-09438-f005:**
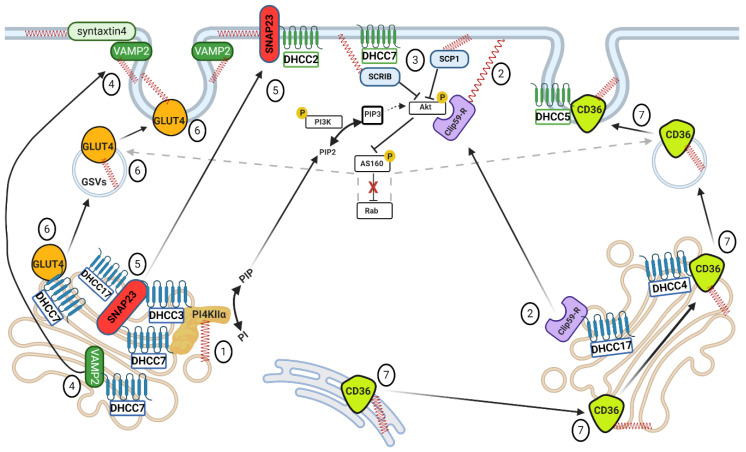
Overview of palmitoylation within insulin signaling and substrate uptake. (1) Palmitoylation of PI4KIIα by DHHC3/7 allows the enzyme to convert PI in PIP, which is the precursor of PIP2. (2) Palmitoylation of Clip59-R by DHHC17 at the Golgi is important for further plasma membrane localization of the protein. Palmitoylated Clip59-R interacts with Akt kinase favoring its activation and downstream signaling (AS160 inhibiting phosphorylation– Rab activation– stimulation of the formation of GLUT4/CD36 vesicles). (3) Palmitoylation of SCP1 and SCRIB (SCRIB palmitoylated by DHHC7) makes these proteins to localize in the proximity of Akt, finally hindering its phosphorylation. (4) v-SNARE VAMP2 is palmitoylated by the Golgi resident DHHC7, thus allowing VAMP2 to mediate fusion of the GSVs at the plasma membrane. (5) t-SNARE SNAP23 is S-acylated at the Golgi by DHHC3, 7 and 17. Palmitoylation favors SNAP23 migration to the plasma membrane, where SNAP23 is further stabilized by DHHC2. Eventually, SNAP23 mediates the fusion of GSVs through interaction with VAMP2. (6) GLUT4 is palmitoylated by DHHC7 at the Golgi. S-acylation of GLUT4 helps the protein to translocate to the cell surface and to mediate glucose uptake. (7) CD36 palmitoylation by (an) unknown DHHC(s) initiates at the level of ER and is essential for correct CD36 processing and folding. CD36 is further palmitoylated at the Golgi by DHHC4, allowing the protein to traffic to the cell surface where it is further stabilized by DHHC5 via S-acylation, finally aiding lipid uptake. Please note that caveolin palmitoylation has been left out. For this, see Figure 4 (caveolin-1). (Golgi-DHHCs in blue, plasma membrane-DHHCs in green, palmitate moieties represented as spring–red shape).

**Table 1 ijms-21-09438-t001:** Basal palmitoylation on insulin signaling and substrate transporters trafficking.

Palmitoylated Protein	Localization of the Palmitoylated Protein	Palmitoyl Acyl Transferase	Effect of the Basally Palmitoylated Protein on Insulin Signaling and Substrates Uptake	Cell Type
Caveolin-1	Plasma membrane [18]	DHHC7, DHHC21 [86]	(Possible) Negative regulation *Mitigate insulin-stimulated glucose uptake*	Adipocytes, HEK293
Caveolin-2	Plasma membrane [96]	Unknown	Positive regulation *Facilitates phosphorylation of IRS-1 by IR* [96]	Adipocytes
Caveolin-3	Plasma membrane [100]	Unknown	Unknown	Cardiomyocytes
PI4KIIα	Golgi [22,102,103]	DHHC3, DHHC7 [22]	(Possible) Positive regulation *Increases PI4P (PIP2 precursor) content*	COS-7, HeLa
Clip59-R	Plasma membrane [104]	DHHC17 [105]	Positive regulation *Facilitates phosphorylation of Akt kinase* [105]	Adipocytes
RabGAP AS160 [26]	Unknown	Unknown	Unknown	Adipocytes
SCRIB	Plasma membrane [106]	DHHC7 [106]	(Possible) Negative regulation *Inhibits PI3K/Akt signaling* *in response to EGF* [106]	HEK293, MCF10A
SCP1	Plasma membrane [107]	Unknown	Negative regulation *Inhibits Akt phosphorylation* *in response to insulin* [107]	HEK293, MEF
VAMP2 (v-SNARE)	GLUT4 vesicles [108]	DHHC7 [108]	Positive regulation *Mediate GLUT4-mediated glucose uptake* [108]	Adipocytes
SNAP23 (t-SNARE)	Plasma membrane [109,111]	DHHC 2,3,7 and 17 [110]	(Possible) Positive regulation *Might increase insulin-stimulated glucose uptake*	PC12, COS-7
IRAP	GLUT4 vesicles [108]	DHHC7 [108]	(Possible) Positive regulation *Mediate insulin-stimulated glucose uptake* [108]	Adipocytes
MUNC18	Unknown	Unknown	Unknown	Adipocytes
GLUT4	Plasma membrane [108,120]	DHHC7 [108]	Positive regulation *Necessary for insulin-stimulated* *stimulated glucose uptake* [108]	Adipocytes
CD36	Plasma membrane [126,127,132]	DHHC4, DHHC5 [127,129]	Positive regulation *Necessary for fatty acids uptake* [127,129]	Adipocytes, COS7

Red color highlights an inhibitory effect of palmitoylation on insulin signaling and substrates uptake; Green color highlights a stimulatory effect of palmitoylation on insulin signaling and substrates uptake.

**Table 2 ijms-21-09438-t002:** Aberrant protein palmitoylation driving insulin resistance.

Palmitoylated Protein	Localization of the Aberrantly Palmitoylated Protein	Effect of the Aberrant Palmitoylation	Cell Type
GAPDH	Cell membranes [133]	Decreases GAPDH enzymatic activity and glucose utilization [133]	Rabbit muscle
PKCε	Actin filament [36]	Increases phosphorylation of S-Acylated PKCε, leading to downregulation of IR transcription [36]	Skeletal muscle Adipocytes
GLUT4	Unknown	Might impair GLUT4 trafficking to the plasma membrane [26]	Adipocytes
CD36	Increased presence at the plasma membrane [28]	Increases both CD36 translocation to the plasma membrane and fatty acids uptake, leading to NASH [28]	Hepatocytes

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
