# Peer review of "Putative Role of Protein Palmitoylation in Cardiac Lipid-Induced Insulin Resistance"

_ijms, 2020, doi:10.3390/ijms21249438_

Round 1

Reviewer 1 Report

The submitted manuscript eas read with care and some concerns are highlighted below: 

The overall manuscript includes so many details of molecular mechanisms and molecular interaction. This is obviously a sign of a detailed work however, on the other hand, it makes reading the manuscript very difficult. In addition to the provided figures, authors might benefit from tabulating the information in a table format to compile the data. 

Line 48-49 mentions palmitoylation in a reversible fashion. I would expect to see more focus on the conclusion part regarding the reversible pattern of palmitoylation on diagnosis or treatment. The conclusion should be expanded in this perspective. 

The legend of Figure 2 shows significant similarities with the group's other paper "Understanding the distinct subcellular trafficking of CD36 and GLUT4 during the development of myocardial insulin resistance". Paraphrasing that part would keep the authors on the safe side in order not to be considered as a self-citation.

Author Response

Reply to the comments raised by reviewer 1:

We wish to thank this reviewer for having taken the time to evaluate our manuscript and for forwarding constructive comments. Below, please find a point-by-point response.

Comments:

  1. The overall manuscript includes so many details of molecular mechanisms and molecular interaction. This is obviously a sign of a detailed work however, on the other hand, it makes reading the manuscript very difficult. In addition to the provided figures, authors might benefit from tabulating the information in a table format to compile the data.
  • Reply:

We understand this comment and, following the reviewer’s suggestion, have added two tables to provide an overview of the palmitoylated proteins involved in insulin signaling and the PATs/APTs involved (see page 17).

  1. Line 48-49 mentions palmitoylation in a reversible fashion. I would expect to see more focus on the conclusion part regarding the reversible pattern of palmitoylation on diagnosis or treatment. The conclusion should be expanded in this perspective.
  • Reply:

We agree that the reversibility of palmitoylation is a unique characteristic that deserves more explicit attention in the “Conclusions and perspectives” section. Therefore, in this section we have added the phrase “In particular, the reversible nature of the palmitoylation process might offer advantages in the diagnosis and treatment of lipid-induced insulin resistance in heart and other tissues.” (Lines 663-665). The lines immediately below then continue with the possible treatment strategies.

  1. The legend of Figure 2 shows significant similarities with the group's other paper "Understanding the distinct subcellular trafficking of CD36 and GLUT4 during the development of myocardial insulin resistance". Paraphrasing that part would keep the authors on the safe side in order not to be considered as a self-citation.
  • Reply:

We thank this reviewer for alerting us to this oversight. We have rephrased this legend in order to not to plagiate ourselves. We appreciate this suggestion of the reviewer, i.e., to keep on the safe side.

Reviewer 2 Report

Schianchi and colleagues review the process of palmitoylation in insulin resistance. The manuscript is well written and deals with an issue of growing importance.

  1. The title of the paper seems inappropriate. Insulin-induced insulin resistance occurs, but very little of this paper deals with this condition. As the authors know, there are many putative mediators of insulin resistance, and many of the examples they cite may be related to these other mediators, such as hyperglycemia. I would suggest that the authors remove the phrase “insulin-induced” from the title. It may be more appropriate to mention “cardiac” in the title since this appears to be the focus of the piece.
  2. CD36 is an important protein in the scheme proposed. The authors should include mention of the recent publication by Jian-Wei Hao et al. demonstrating that dynamic palmitoylation of CD36 regulates endocytosis—Nature Communications 11:4765 (21 Sept, 2020).
  3. On line 127 (in the legend to Figure 1), GSV is referred to as glucose storage vesicle but this is incorrect. GSV represents GLUT4 storage vesicle.
  4. Figure 5 needs to be modified. The placement of the palmitate moieties is erratic, with palmitate sometimes not even contacting the modified protein. DHHC PATs are represented in a very misleading manner. DHHCs are transmembrane proteins, and yet they are drawn vaguely, including an image of DHHC2 that implicates the presence of this enzyme in the extracellular space.
  5. The final section needs to be modified. Lines 621-622 states that the redundance of APTs does not make these enzymes the right candidates to modulate depalmitoylation of specific proteins. If this reasoning is followed, DHHC enzymes are much worse to pursue since there are 23 DHHCs and only 2 APTs. Lines 631-632 acknowledge that PATs show some level of redundancy, but this is misleading. Most tissues contain multiple DHHC isoforms, again making this line of thinking confusing.
  6. Another concept in the final section is misleading. There is the statement on lines 628-629 that improved glucose influx into cardiomyocytes would lead to re-acidification of endosomes, and while this might be true, this strategy might be harmful in the failing heart, which is relevant to obesity and diabetes. Failing hearts shift their metabolism from preferential use of fatty acids to glucose, which is thought to be a maladaptive change, which is not considered by this notion of increasing glucose uptake in cardiomyocytes.

Author Response

Reply to the comments raised by reviewer 2:

We thank this reviewer for her/his praising words and for forwarding constructive comments. Below, please find a point-by-point response.

Comments:

  1. The title of the paper seems inappropriate. Insulin-induced insulin resistance occurs, but very little of this paper deals with this condition. As the authors know, there are many putative mediators of insulin resistance, and many of the examples they cite may be related to these other mediators, such as hyperglycemia. I would suggest that the authors remove the phrase “insulin-induced” from the title. It may be more appropriate to mention “cardiac” in the title since this appears to be the focus of the piece.
  • Reply:

We sincerely apologize, the term “insulin-induced” was a mistake and should be “lipid-induced”. We thank this reviewer for alerting us. We have changed the title now, and also included the term “cardiac”.

  1. CD36 is an important protein in the scheme proposed. The authors should include mention of the recent publication by Jian-Wei Hao et al. demonstrating that dynamic palmitoylation of CD36 regulates endocytosis—Nature Communications 11:4765 (21 Sept, 2020).
  • Reply:

We have now included this reference (nr. 129) and described these novel findings in section 4.7 on CD36 (lines 501-505).

  1. On line 127 (in the legend to Figure 1), GSV is referred to as glucose storage vesicle but this is incorrect. GSV represents GLUT4 storage vesicle.
  • Reply:

We thank the reviewer for alerting us to this error, which now has been corrected.

  1. Figure 5 needs to be modified. The placement of the palmitate moieties is erratic, with palmitate sometimes not even contacting the modified protein. DHHC PATs are represented in a very misleading manner. DHHCs are transmembrane proteins, and yet they are drawn vaguely, including an image of DHHC2 that implicates the presence of this enzyme in the extracellular space.
  • Reply:

We agree that the figure could be improved. Thus, we have modified this figure according to this reviewers’ suggestions, using distinct representations for the DHHCs. Consequently, the figures 3 and 4 have been modified accordingly.

  1. The final section needs to be modified. Lines 621-622 states that the redundance of APTs does not make these enzymes the right candidates to modulate depalmitoylation of specific proteins. If this reasoning is followed, DHHC enzymes are much worse to pursue since there are 23 DHHCs and only 2 APTs. Lines 631-632 acknowledge that PATs show some level of redundancy, but this is misleading. Most tissues contain multiple DHHC isoforms, again making this line of thinking confusing.
  • Reply:

We modified text to better clarify this issue (lines 672-679 and lines 685-687). We also removed the term “redundancy” in this context because it may lead to misinterpretation indeed.

  1. Another concept in the final section is misleading. There is the statement on lines 628-629 that improved glucose influx into cardiomyocytes would lead to re-acidification of endosomes, and while this might be true, this strategy might be harmful in the failing heart, which is relevant to obesity and diabetes. Failing hearts shift their metabolism from preferential use of fatty acids to glucose, which is thought to be a maladaptive change, which is not considered by this notion of increasing glucose uptake in cardiomyocytes.
  • Reply:

The reviewer is indeed correct that such treatment might be not beneficial in the context of end-stage heart failure, in which the heart might benefit from every substrate that it can access. However, this treatment targeting PATs is aimed at restoring lipid accumulation in the earlier stages of cardiac dysfunction, in which there is not yet extensive fibrosis. We have now more clearly indicated that this treatment is of potential use in the pre-diabetic heart (see lines 683-684).